# Homotopy Learning of Parametric Solutions to Constrained Optimization Problems

## Abstract

Building deep learning (DL) alternatives to constrained optimization problems has been proposed as a cheaper solution approach than classical constrained optimization solvers. However, these approximate learning-based solutions still suffer from constraint violations. From this perspective, reaching a reliable convergence remains an open challenge to DL models even with state-of-the-art methods to impose constraints, especially when facing a large set of nonlinear constraints forming a non-convex feasible set. In this paper, we propose the use of homotopy meta-optimization heuristics which creates a continuous transformation of the objective and constraints during training, to promote a more reliable convergence where the solution feasibility can be further improved. The method developed in this work includes 1) general-purpose homotopy heuristics based on the relaxation of objectives and constraint bounds to enlarge the basin of attraction and 2) physics-informed transformation of domain problem leading to trivial starting points lying within the basin of attraction. Experimentally, we demonstrate the efficacy of the proposed method on a set of abstract constrained optimization problems and real-world power grid optimal power flow problems with increasing complexity. Results show that constrained deep learning models with homotopy heuristics can improve the feasibility of the resulting solutions while achieving near-optimal objective values when compared with non-homotopy counterparts.

## 1 Introduction

Recent years have seen a rich literature of deep learning (DL) models for solving constrained optimization problems on real-world tasks such as power grid, traffic, or wireless system optimization. These applications can largely benefit from data-driven alternatives enabling fast real-time inference. The problems remain that these problems commonly include a large set of nonlinear system constraints that lead to non-convex parametric nonlinear programming (pNLP) problems which are NP-hard. Earlier attempts simply adopt imitation learning (i.e., supervised learning) to train function approximators via a minimization of the prediction error using labeled data of pre-computed solutions. Unfortunately, these models can hardly perform well on unseen data as the outputs are not trained to satisfy physical constraints, leading ifeasible solutions.

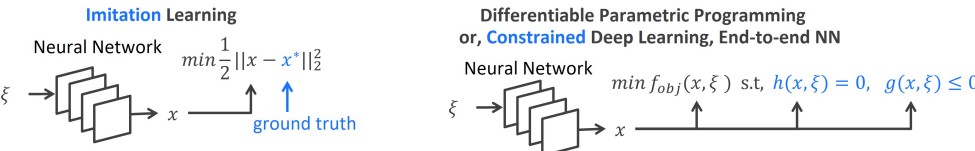

Figure 1: Imitation learning VS end-to-end learning using Differentiable Parametric Programming

To address the feasibility issues, existing methods have explored the imposing of constraints on the output space of deep learning models. Section 2 provides an overview of the existing techniques. The imposing of constraints has inspired the use of end-to-end learning approaches that directly consider the original objectives and constraints in the NN training process without the need of expert labeled data.

However, even the state-of-the-art methods to impose constraints can hardly guarantee a reliable convergence with perfect feasibility on unseen data for large problems. Penalty method which treats the constraints as a form of regularization requires careful selection of penalty weights and such soft-constraint treatment cannot guarantee satisfying constraints to machine precision. Primal-dual Lagrangian-based formulation theoretically provides a hard constraint methodology, whereas empirical evidence indicate it can perform worse than penalty method (the reason remains unclear, see (Márquez-Neila et al., 2017)). Another strategy (Donti et al., 2021) adds *completion layer* after the NN model to reconstruct the complete solution from an incomplete one given by the NN, using the equality constraints. This enables a hard constraint method for equality constraints, whereas when facing nonlinear constraints, the completion layer, as an iterative solver, adds to the computation complexity, and can potentially diverge when a bad incomplete output from NN causes a non-existence of feasible solution to be reconstructed.

Due to the lack of consensus in the community, these new approaches are often called by different names such as constrained deep learning, end-to-end neural networks, differentiable optimization layers, or deep declarative networks. In this paper we contribute to this diversity by referring to the proposed method as differentiable parametric programming (DPP) to emphasize the connection with sensitivity analysis developed in the context of operations research (Gal & Nedoma, 1972; Gal & Greenberg) and later adopted in control theory applications (Bemporad et al., 2000; Herceg et al., 2013).

As a main contribution, we present a novel method that combines homotopy, deep learning and parametric programming formulations into one coherent algorithmic framework. The aim is to obtain a more reliable convergence of constrained deep learning models whose solution feasibility can be further improved. Homotopy based meta-optimization heuristics are developed to create a continuous transformation of objective and constraint sets, making a homotopy path that drives the training of NN to gradually learn from easy problems to harder problems. Our contribution includes 2 types of homotopy heuristics which are different ways of utlizing the basin of attraction: 1) homotopy heuristics based on relaxation of objective and constraints to manipulate the basin of attraction, 2) domain-aware homotopy heuristics based on physics-informed transformation of the problem to make it available trivial starting points within the basin of attraction

## 2  RELATED WORK

### 2.1  CONSTRAINED NEURAL NETWORKS

Imposing constraints onto the output space of NNs can be done via supervised learning (where labels are used to write the constraints) or unsupervised learning; using either soft constraint (which usually treats the constraints as a regularization) or hard constraint method (which usually means enforcing satisfaction of constraints to machine precision, i.e., perfect satisfaction). We briefly describe the different categories of existing methods, according to the type of constraints to be imposed:

**General equality and inequality constraints** can be imposed by augmented objective functions, reprojection (as hard constraints), completion layer (as hard constraints), etc. Among augmented objective function methods, penalty method (Yang et al., 2019; Hu et al., 2020; Donti et al., 2021; Pan et al., 2019) augments the objective function by additional terms that penalize the violation of constraints, treating the constraints as a regularization with pre-defined weights to control the regularization strength, whereas the primal-dual based formulation (or lagrangian formulation) (Nandwani et al., 2019; Fioretto et al., 2020; Márquez-Neila et al., 2017) exploits Lagrangian duality and iteratively updates both primal and dual variables to minimize a Lagrangian loss function. Penalty method, as a soft constraint method, has some theoretical deficits of requiring extra weight tuning for the multi-objective loss function, and no guarantee of satisfying constraints. However, evidence (Márquez-Neila et al., 2017) has shown that Lagriangian formulation, as a hard constraint method, is empirically worse. Reprojection method makes corrections on out-of-constraint-set outputs by projecting them onto the feasible region, either during the training cycle using different variants of projected gradient descent methods(Donti et al., 2021; Márquez-Neila et al., 2017), or during the test as a post-processing step (e.g.,(Pan et al., 2019) passed outputs to a physical equation solver). A completion layer method (e.g., DC3(Donti et al., 2021), ACnet(Beucler et al., 2021)) developed NN to only produce a subset of the target output variables, and then an extra constraint layer attached

after NN computes the remaining outputs according to constraints. These methods have pros and cons, as discussed in Section 1.

**Domain-specific constraints**. Some real-world applications work with graphical structures, necessitating the encoding of network topology constraints i) in model architecture as a hard constraint (e.g., Graph Neural Network (GNN) (Kundacina et al., 2022; Donon et al., 2019; Owerko et al., 2020; Diehl, 2019) and other graphical models(Li et al., 2022)), ii) in prior as soft constraint (e.g, adjacent matrix as prior (Yang et al., 2019; Hu et al., 2020) , or iii) in input features (topology related NN inputs). More attempts to impose dynamic and recursive constraints include i) unrolled neural networks (soft constraint) where recurrent neural network (RNN) and its variants unroll differentiable dynamic models (Tuor et al., 2022)(Skomski et al., 2021)(Drgoňa et al., 2021) and iterative physical solvers (Zhang et al., 2019)(Yang et al., 2020; Zhang et al., 2018), or ii) encoding temporal and spatial constraints in latent representations (Yuan et al., 2021).

## 2.2 CONSTRAINED OPTIMIZATION

Consider a general constrained optimization problem, with $\xi$ denoting the known parameters representing input data, and $x$ denoting the solution of the corresponding optimization problem. Given a parameter instance $\xi$, the aim is to obtain optimal $x$ by solving:

$$\min_x f_{obj}(x, \xi) \quad \text{s.t} \quad g(x, \xi) \leq 0, \quad h(x, \xi) = 0 \tag{1}$$

When there are non-linear objectives or constraints, (1) defines a family of parametric non-linear programming (pNLP) problems. pNLP problems are NP-hard and handled poorly with state-of-the-art optimization solvers due to the non-convexities in the solution space. Existing works have explored a large number of techniques (Wächter & Biegler, 2005; 2006; Byrd et al., 2000; Liao, 2004), including filter method, line-search, corrections, trust region method, or homotopy methods, to improve on (local) convergence, and also developed many heuristics to allow faster performance of online optimization solvers. The local (and global) convergence properties of optimization methods have also been extensively studied in order to develop tools to improve on convergence guarantee for nonlinear programming. However, scalable solution approaches for generic pNLP problems remain a challenge. Reasons include the numerical difficulties (caused by uncertain behaviors of initialization procedures, stopping criteria, etc), ill-conditioning, and the assumptions required to ensure convergence becoming easily violated in practice.

## 2.3 POWER GRID OPTIMIZATION PROBLEM

One domain-specific problem of interest to this paper is the AC optimal power flow (ACOPF) problem (Capitanescu, 2016)(Pandey et al., 2020) which is hard to solve due to non-convexities. ACOPF is the fundamental optimization problem to determine the optimal control of generator output that can meet the demand with maximal cost-efficiency while safely operating the system within its technical limits. A simple definition of ACOPF is given below, where we minimize the generation cost subject to power balance equations and variable bounds:

$$\min_{x=[V^{real}, V^{imag}, P_g, Q_g]} \sum_{i=1}^{ng} \alpha_i P_{gi}^2 + \beta_i P_{gi} + \gamma_i \tag{2a}$$

$$\text{s.t.} \tag{2b}$$

$$\text{Power balance: } (P_g - P_d) + j(Q_g - Q_d) = V \odot (Y_{bus}V)^* \tag{2c}$$

$$\text{reference bus angle: } V_{ref}^{imag} = 0 \tag{2d}$$

$$\text{voltage magnitude bounds: } |V|_i^- \leq |V|_i \leq |V|_i^+, \quad i \in nb \tag{2e}$$

$$\text{Pgen bounds: } P_{gi}^- \leq P_{gi} \leq P_{gi}^+, \quad i \in ng \tag{2f}$$

$$\text{Qgen bounds: } Q_{gi}^- \leq Q_{gi} \leq Q_{gi}^+, \quad i \in ng \tag{2g}$$

where $P_d, Q_d$ are unknown parameters of load demand (given as input information), $\alpha_i, \beta_i, \gamma_i$ are generator cost coefficients of the i-th generator, $P_{gi}, Q_{gi}$ denote real and reactive power output of the i-th generator, $V^{real}, V^{imag}$ denote vector of real and imaginary voltage at all buses, $V$ is a vector of complex bus voltages $V = V^{real} + jV^{imag}$, $|V|$ denotes the magnitude of voltage, and $x^+, x^-$ denote the upper and lower variable bound.

## 2.4 Homotopy Method for Constrained Optimization

Homotopy method is a type of meta-heuristics to handle hard problems which can otherwise easily divergence or converge to a bad point. It decomposes the original (nonlinear) problem $F(x)$ to a series of sub-problems $H(x, \lambda_H)$, creating a path of optimizers driven by the change of homotopy parameter $\lambda_H$. As $\lambda_H$ shifts from 0 to 1, the sub-problem $H(x, \lambda_H)$ continuously transforms from a simple problem to the original one. Commonly, homotopy-based heuristics have been used for local optimization of nonlinear problems. The most popular form of designing $H(x, \lambda_H)$ is via a linear combination of a trivial problem $H_0(x)$ and the original one such that $H(x, \lambda_H) = (1 - \lambda_H)H_0(x) + \lambda_H F(x)$. Existing works have developed perturbation techniques for general nonlinear problems (He, 1999; Liao, 2004), multi-objective problems(Hillermeier et al., 2001) , etc.

Domain-specific homotopy heuristics have also been developed to design $H(x, \lambda_H)$ whose solutions are trivial when $\lambda_H = 0$. For example, in the domain of circuit simulation (Najm, 2010), Gmin-stepping initially shorts all nodes to ground and gradually remove the short-circuit effect; Tx-stepping initially shorts all transmission lines, and then gradually returns to the original branches. Works in (Pandey et al., 2018)(Pandey et al., 2020)(Jereminov et al., 2019) further applied the circuit-theoretic homotopy ideas to power grid simulation and optimization tasks. On the other hand, some works (Dunlavy & O'Leary, 2005) also extended homotopy methods to global optimization tasks through an ensemble of solution points at each homotopy step, and investigated the probability bound on the convergence to global minimizer. Whereas, many homotopy methods are empirically slower than most other convergence heuristics (e.g., line search, trust region method, etc). Therefore, the use of homotopy on classical optimization solvers can still suffer from limited time efficiency.

## 3 Homotopy Learning for Differentiable Parametric Programming

Here we present a novel method that combines homotopy, deep learning and parametric programming in one coherent algorithmic framework. Specifically, using neural network as an approximation of constrained optimization solvers allows fast real-time inference for any new input, and meanwhile, integrating homotopy into the training process facilitates the NN to reach a more reliable convergence with improved feasibility.

### 3.1 Differentiable Parametric Programming

To deal with the challenge regarding scalability in generic pNLP problems, as well as the drawbacks in imitation learning, we can build a data-driven alternative to the traditional optimization solver, with the target objectives and constraints directly integrated within the training cycle. This can be achieved by differentiable parametric programming, which adopts an unsupervised learning of NN model, mathematically defined as:

$$\min_{\Theta} f_{obj}(x, \xi) \text{ s.t. } g(x, \xi) \leq 0, \ h(x, \xi) = 0, \ x = \pi_{\Theta}(\xi), \forall \xi \in \Xi \tag{3}$$

with $\pi_{\Theta}$ denoting a NN model mapping from input $\xi$ to the output solution $x$, and $\Theta$ being the NN weights. Figure 1 illustrates the difference from imitation learning.

One methodology of interest to this paper is the penalty method, which imposes the constraints by reformulating (3) into an unconstrained form, leading to a NN loss function as below:

$$\min_{\Theta} f_{obj}(\pi_{\Theta}(\xi), \xi) + \sum_i w_i \cdot P_i(h_i(\pi_{\Theta}(\xi), \xi)) + \sum_i w_i \cdot P_i(g_i(\pi_{\Theta}(\xi), \xi)) \tag{4}$$

with $P_i()$ penalizing the violation of constraints, and hyperparameters $w$ denoting the pre-defined penalty weights. Popular selections of $P_i()$ include residual norm penalty, typically L2-norm, for equality constraints and ReLU operator for inequality constraints. Beyond these popular forms, work in (Zhu et al., 2019) further explored using variational functionals of partial differential equations (PDE) as penalty terms to impose PDE constraints.

### 3.2 Method Overview and Theoretical Foundations

As discussed earlier, when facing non-convex objective function and constraints in (3), even the state-of-the-art methods can have difficulty reaching a good converge with small violation of constraints. In this paper we develop a novel optimization heuristic for these type of problems based on the idea of homotopy (He, 1999; Liao, 2004).

Specifically, we apply homotopy to the problem by creating a continuous transformation of the objecitve and constraints in equation 3, such that a subproblem in the homotopy path can be expressed as:

$$\min_{\Theta} f_{\lambda_H}(x, \xi) \text{ s.t. } g_{\lambda_H}(x, \xi) \leq 0, \ h_{\lambda_H}(x, \xi) = 0, \ x = \pi_{\Theta}(\xi), \ \forall \xi \in \Xi \tag{5}$$

As the homotopy parameter $\lambda_H$ incrementally changes from 0 to 1, the objective and constraints $f_{\lambda_H}, g_{\lambda_H}, h_{\lambda_H}$ gradually return back to $f_{obj}, g, h$, shifting the task from an easy-to-solve problem to the original one. Driven by this transformation, the neural network model $\pi_{\Theta}$ gradually learns to approximate the solution of harder and harder problems.

Next we describe the conceptual idea of our homotopy heuristics along with some theoretical foundations. Let $H(x, \lambda_H)$ denote a sub-problem in the homotopy path that transforms the original optimization problem $F(x)$ in (3). We can use a certain local minimization method as the solver of $H(x, \lambda_H)$ to get $x_{\lambda_H}^*$, a local minimizer of $H(x, \lambda_H)$. For that local minimization method, we can define the *basin of attraction* $\mathcal{B}(\lambda_H)$ of a local minimizer of $H(x, \lambda_H)$ as the set of points $x$ such that the local minimization method started at $x \in \mathcal{B}(\lambda_H)$ will converge to that minimizer $x_{\lambda_H}^*$ (Dunlavy & O'Leary, 2005).

Assuming that the original problem $F(x)$ has a global minimizer $x^*$ that is unique and *isolated*. According to studies in (Dunlavy & O'Leary, 2005), the Implicit Function Theorem guarantees that there exists of unique curve of isolated minimizers that passes through $(x^*, 1)$, meaning the desirable optimum is theoretically accessible through a curve of homotopy problem minimizers $(x_{\lambda_H}^*, \lambda_H)$.

Importantly, when taking small enough homotopy step, i.e., $\lambda_H^{(k)} = \lambda_H^{(k-1)} + \Delta\lambda_H$ with small enough $\Delta\lambda_H$, there is a high likelihood that the minimizer of a subproblem $H(x, \lambda_H^{(k-1)})$ is in the *basin of attraction* of the next subproblem $H(x, \lambda_H^{(k)})$, i.e., $x^{*(k-1)} \in \mathcal{B}(\lambda_H^{(k)})$. This motivates us to carefully design $H(x, \lambda_H)$ and select $\Delta\lambda_H$ to generate a easy homotopy path where, starting with a proper $H(x, 0)$, each following sub-problem $H(x, \lambda_H^{(k)})$ is solved without difficulty due to a good starting point given from $H(x, \lambda_H^{(k-1)})$. Figure 2 illustrates such a homotopy path.

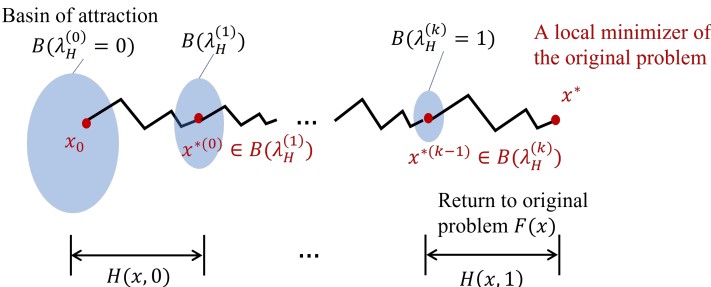

Figure 2: A desirable theoretical homotopy path that gradually leads to a minimizer of the original problem: each subproblem is easily solvable by starting from a point in the basin of attraction $\mathcal{B}(\lambda_H)$

Based on these findings, the main idea of our homotopy heuristics include:

1. **creating proper transformation of the objective and constraints** via a linear combiniation $H(x, \lambda_H) = (1 - \lambda_H)H_0(x) + \lambda_H F(x)$, with the initial problem $H(x, \lambda_H = 0) = H_0(x)$ being easily solvable. The illustration in Figure 2 reveals 2 different ways to design $H_0(x)$:

- **TYPE I**: expanding the basin of attraction $\mathcal{B}(\lambda_H)$ to a larger volume in the initial homotopy step $H_0(x)$. In this case, the initial problem is easy to solve in a way that there exist more

choices of starting point that can enable solution trajectories to a (local) minimizer. The basin of attraction gradually shrinks it as $\lambda_H$ increases.

- **TYPE II**, designing $H_0(x)$ such that a trivial starting point $x_0 \in \mathcal{B}(\lambda_H = 0)$ is available (before any training starts). In this case, $H_0(x)$ is easy in a way that we have a starting point within its basin of attraction, which guarantees a solution trajectory to a minimizer.

2. **selection of a proper homotopy step** $\Delta\lambda_H$ that trades off between the likelihood of $x^{*(k-1)} \in \mathcal{B}(\lambda_H^{(k)})$, and an acceptable time complexity.

Eventually, the overall training process can be described as Algorithm 1:

---
**Algorithm 1** Homotopy Learning of Neural Network
---
1: **Initialize**: homotpy parameter $\lambda_H \leftarrow 0$, NN model $\pi_\Theta$
2: **while** $\lambda_H \leq 1$ **do**
3:     **1. Update** objective and constraints $f_{\lambda_H}, h_{\lambda_H}(\cdot) = 0, g_{\lambda_H}(\cdot) \leq 0$
4:     **2. Train NN** using penalty method
5:     **while** not converged **do** for all $\xi \in \Xi$:
6:         Forward pass: $x \leftarrow \pi_\Theta(\xi)$
7:         Get loss: $L_{\lambda_H} \leftarrow f_{\lambda_H}(x) + \sum_i w_{eq} \cdot ||(h_{\lambda_H}(x))||^2 + \sum_i w_{ineq} \cdot relu(g_{\lambda_H,i}(x))$
8:         Backward pass: $\pi_\Theta \leftarrow \arg\min_\Theta L_{\lambda_H}$
9:     **end while**
10:    **3. Update** homotpy parameter $\lambda_H \leftarrow \lambda_H + \Delta\lambda_H$
11: **end while**
---

### 3.3 TYPE I: RELAXATION BASED HEURISTICS FOR OBJECTIVE AND CONSTRAINTS

Let's first focus on TYPE I homotopy methods which enlarge the basin of attraction to create easily solvable problems at the initial homotopy steps. We propose the use of relaxation based heuristics, which work by convexifying the objective and expanding the feasible constraint set at the beginning of the homotopy process.

We briefly illustrate the idea behind using relaxation. Mathematically, consider minimizing a non-convex smooth objective function $f_{obj}$ subject to a constraint set $\mathcal{S}$, using penalty method (or barrier method, etc) to reformulate the problem. Let $L$ denote the (smooth) augmented loss function, $x^*$ denote one of the (local) minimizers $x^* \in \mathcal{S}$, and let $\mathcal{N}(x^*)$ denote a local neighborhood of $x^*$ where $L$ is a generic (Lipschitz smooth) convex function. Now consider using gradient descent as the local optimization method with its learning rate lower than twice the smallest optimal learning rate for any component ($\eta < 2\min\eta_{i,opt}$, studies have shown that otherwise learning will diverge), then starting from any $x \in \mathcal{N}(x^*)$, we can guarantee to reach $x^*$ as the final solution. Therefore, $\mathcal{N}(x^*)$ can be considered a lower bound of the basin of attraction for $x^*$, i.e., $\mathcal{B}(x^*) \supseteq \mathcal{N}(x^*)$.

As we relax (convexify) the objective function, the resulting augmented loss function $L^+$ has higher convexity, making the optimization landscape more convex. Then for a certain local minimizer $x^*$, there's a likelihood that $x^*$ has a larger neighborhood $\mathcal{N}^+(x^*)$ where $L^+$ is generic convex, in which case the lower bound of $\mathcal{B}(x^*)$ increases:

$$\mathcal{N}^+(x^*) \supseteq \mathcal{N}(x^*) \tag{6}$$

Further when relaxing the constraint set from $\mathcal{S}$ to $\mathcal{S}^+$, with $\mathcal{S} \subseteq \mathcal{S}^+$ and let $\chi^*(\mathcal{S})$ denote the set of all local minimizers in the constraint set $\mathcal{S}$, then the expanded constrained set is likely to contain more local minimizers, i.e.,

$$\chi^*(\mathcal{S}^+) = \chi^*(\mathcal{S}) \cup \chi^*(\mathcal{S}^+ \setminus \mathcal{S}) \supseteq \chi^*(\mathcal{S}) \tag{7}$$

Therefore the total basin of attraction (the set of starting points that will lead to any local minimizer), as the union of $\mathcal{B}(x^*)$ for each individual minimizer $x^*$, will expand after relaxation:

$$B_{total}^+ = \cup_{x^* \in \chi^*(\mathcal{S}^+)}\mathcal{B}(x^*) \supseteq \cup_{x^* \in \chi^*(\mathcal{S}^+)}\mathcal{B}(x^*) = B_{total} \tag{8}$$

Below we introduce the relaxation heuristics. To handle non-convex objective functions during the homotopy process, we apply the heuristics of Convexify Objective (CObj):

**Convexify Objective (CObj)**: Given a non-convex objective function $f_{obj}$, the homotopy path starts from minimizing a convex objective function $f_{cvx}$ (which is close to the original $f_{obj}$), and gradually insert non-convexity via a linear combination with the original objective:

$$f_{\lambda_H} = \lambda_H * f_{obj} + (1 - \lambda_H)f_{cvx} \tag{9}$$

For inequality constraints $g(\cdot) \leq 0$ during the homotopy process, we propose 2 heuristics:

**1) Shrink Bounds (SBnds)**: In the homotopy process, the constraints transforms by $g \leq (1 - \lambda_H)\epsilon^+$. As $\lambda_H$ increases from 0 to 1, the inequality constraints gradually transforms from $g \leq \epsilon^+$ to the original constraints $g \leq 0$. This is intuitive for variable bounds $x^- \leq x \leq x^+$ which can be rewritten in the homotopy process as:

$$\lambda_H x^- + (1 - \lambda_H)\epsilon^- \leq x \leq \lambda_H x^+ + (1 - \lambda_H)\epsilon^+ \tag{10}$$

where $\epsilon^+, \epsilon^-$ are some pre-defined relaxation of upper and lower bounds making it easier to satisfy. During the homotopy process, the bounds are gradually tightened as $\lambda_H$ increases.

**2) Grow Penalty (GPen)**: Another homotopy heuristic exploits the penalty strength of the violation of constraints. The transformation of constraints can be expressed as:

$$\lambda_H g \leq 0 \tag{11}$$

such that the increasing $\lambda_H$ leads to a growing penalty of the constraint violations when the constraints are included in the augmented loss function.

Finally, we propose the "Split and Shrink" (SaS) heuristics for equality constraints $h(\cdot) = 0$:

**Split and Shrink (SaS):** Any equality constraint $h(\cdot) = 0$ can be equivalently split into two inequality constraints $0 \leq h \leq 0$. These constraints can be relaxed via a perturbation of the bounds: $-\epsilon \leq h \leq \epsilon$ with $\epsilon > 0$. A larger perturbation $\epsilon$ makes the constraints easier to satisfy. This motivates us to design a homotopy path where the perturbation gradually decreases. In more detail, with a predefined large perturbation $\epsilon_H$ and tiny perturbation $\epsilon_L$, the split constraints are perturbed by:

$$-\epsilon_L - (1 - \lambda_H)\epsilon_H \leq h \leq \epsilon_L + (1 - \lambda_H)\epsilon_H \tag{12}$$

As the homotopy parameter $\lambda_H$ shifts from 0 to 1, the decreasing perturbation leads to a tighter bounds that enforces $h(\cdot) = 0$ more closely.

### 3.4 TYPE II: DOMAIN-AWARE TRANSFORMATION WITH TRIVIAL SOLUTIONS AVAILABLE

This section further explores TYPE II homotopy heuristics which makes it available a trivial starting point within the basin of attraction $\mathcal{B}(\lambda_H = 0)$. Specifically, consider minimizing an objective function $f_{obj}$ subject to a constraint set $\mathcal{S}$, we aim to transform the problem via a carefully designed manipulation of the objective (if non-convex) and the constraint functions, such that the purturbed problem $H(x, \lambda_H = 0) : \min f_{perturbed}, s.t.\mathcal{S}_{purturbed}$ is easy-to-solve in a way that, before any training starts, a trivial starting point $x_0 \in \mathcal{B}(\lambda_H = 0)$ is available to guarantee a solution trajectory towards a minimizer.

Instead of a simple relaxation of the constraint bounds, transforming an entire problem into one which has trivial solutions (or trivial starting points) often requires some domain knowledge. In this paper, we work in the context of power grid, and show the design of 2 domain-specific homotopy heuristics: load-stepping and Tx-stepping to impose highly non-linear equality constraints. We consider the power grid optimization control problem defined in (2) and the power system related symbols used here are also based on the definitions in Section 2.3.

#### 3.4.1 LOAD-STEPPING

The homotopy method of load stepping creates a path of power flow balance constraints induced by a gradual increase of load demand:

$$h_{\lambda_H} = (P_g - \lambda_H * P_d) + j(Q_g - \lambda_H * Q_d) - v \odot (Y_{bus}v)^* = 0 \tag{13}$$

From a domain perspective, when $\lambda_H = 0$, all load demands are zeroed ($\lambda_H P_d = \lambda_H Q_d = 0$) and all variable bounds can be removed/relaxed using the heuristics desgined in Section 3.3 for

inequalities. A trivial solution to this problem exists: $x_0 = [V_r, V_i, P_g, Qg] = \mathbf{0}$, representing that the power grid is closed off everywhere with no supply and demand. We make use of this trivial solution via a warm-homotopy loss:

$$l_{warm} = w_{warm}(x - x_0)^T(x - x_0) \tag{14}$$

to guide the update of the deep learning model towards a quick convergence to a feasible point for (only) the first homotopy step. A physics-informed homotopy optimization using penalty method can therefore be written as $H(x, \lambda_H) : \min L_{\lambda_H} + I_0(\lambda_H) * l_{warm}$ with $I_0(\lambda_H)$ being an indicator function, and $L_{\lambda_H}$ is an augmented loss function as defined in Alogrithm 1.

### 3.4.2 TX-STEPPING

Unlike load stepping which manipulates load demand, Tx-stepping manipulates the branches instead, through a continuous transformation of the bus admittance matrix $Y$:

$$h_{\lambda_H} = (P_g - P_d) + j(Q_g - Q_d) = v \odot (Y_{\lambda_H}v)^* = 0 \tag{15}$$

where $Y_{\lambda_H} = \lambda_H Y_{bus} + (1-\lambda_H)Y_0$. From a domain perspective, at the first homotopy step $\lambda_H = 0$, we can replace all branches with zero-resistance low-impedance lines (e.g. impedance $= 0 + 10^{-4}i$), giving a bus admittance matrix $Y_0$. This creates all nearly shorted branches with nice properties that 1) lossless lines lead to no real power loss during transmission, i.e., $\sum P_g = \sum P_d$, and 2) $v_i \approx v_{ref}$ for any bus $i$, specifically, all bus voltage magnitudes are close to the reference bus values due to the low voltage drops, and all bus voltage angles will lie within a $\epsilon$-small radius around the refernce bus angle. These nice properties make the problem easily solvable and a trivial solution $x_0$ available.

## 4 NUMERICAL RESULTS

We evaluate the efficacy of homotopy heuristics on both general non-convex constrained optimization problems and the real-world problem of power grid optimal power flow. The use of homotopy is expected to enable a more reliable convergence of the neural network models to outperform non-homotopy results on the following criteria:

- **Optimality**: the objective function $f_{obj}$ achieved by the solution. For ACOPF problem it represents the per-hour cost ($/h) of the generation dispatch.
- **Feasibility**: how much the solution $\hat{x}$ violates the equality and inequality constraints. Feasibility is quantified by the mean and maximum violation of constraints: $\text{mean}(h(\hat{x})), \max(h(\hat{x})), \text{mean}(relu(g(\hat{x}))), \max(relu(g(\hat{x})))$. In real-world tasks, smaller violation of constraints means the neural network outputs a more practical solution for real-world optimization and control.

We compare the different versions of homotopy optimization with non-homotopy settings. The non-homotopy baseline is a vanilla penalty method as formulated in (4), whereas the homotopy settings are combinations of different heuristics added to the vanilla method. Appendix A describes the details on experiment settings and hyper-parameter tuning, in order for a fair comparison.

### 4.1 NON-CONVEX OPTIMIZATION WITH RANDOM LINEAR CONSTRAINTS

First consider a problem with non-convex objective and random linear inequality constraints:

$$\min_x \sum_{i=1}^{n-1} (1 - x_i)^2 + 2(x_{i+1} - x_i^2)^2 \quad \text{s.t.} \quad Ax \le b + C\xi \tag{16}$$

$n$ is the problem size (complexity), $x$ is the solution vector representing $n$ variables to solve, $\xi$ is an $round(0.4 * n) \times 1$ vector representing the (known) input parameter that varies across instances, $A^{n \times n}, b^{n \times 1}, C^{dim(\xi)}$ are randomly generated matrices representing $n$ random linear constraints.

Table 1 shows results on test data for problems with varying complexity. Results demonstrate that adding homotopy heuristics onto the penalty method enables neural network outputs to have a smaller violation of constraints, which means better feasibility. See Appendix A for details on experiment settings.

| Method | | 5 ($\Delta\lambda_H = 0.05$) | Complexity $n$ 25 ($\Delta\lambda_H = 0.05$) | 50 ($\Delta\lambda_H = 0.05$) | 100 ($\Delta\lambda_H = 0.005$) $w_{ineq} = 500$ |
|---|---|---|---|---|---|
| **CObj, SaS, SBnds** | obj mean viol max viol | 0.48 0.0000(0.0000) 0.0000(0.0000) | 31.71 0.0001(0.0002) 0.0016(0.0048) | 62.03 0.0003(0.0011) 0.0129(0.0526) | 276.16 0.0002(0.0009) 0.0246(0.0947) |
| **CObj, SaS, GPen** | obj mean viol max viol | 0.48 0.0000(0.0000) 0.0000(0.0000) | 31.69 0.0001(0.0005) 0.0029(0.0121) | 62.59 0.0002(0.0005) 0.0061(0.0141) | 253.42 0.0004(0.0018) 0.0297(0.1032) |
| **SaS, SBnds** | obj mean viol max viol | 0.48 0.0000(0.0000) 0.0000(0.0000) | 33.35 0.0001(0.0003) 0.0023(0.0066) | 62.33 0.0002(0.0009) 0.0082(0.0222) | 272.36 0.0002(0.0009) 0.0180(0.0711) |
| **SaS, GPen** | obj mean viol max viol | 0.47 0.0000(0.0000) 0.0000(0.0000) | 32.15 0.0001(0.0004) 0.0024(0.0080) | 62.20 0.0006(0.0035) 0.0095(0.0347) | 447.32 0.0031(0.0067) 0.1990(0.3583) |
| **Vanilla** (penalty method) | obj mean viol max viol | 0.46 0.0000(0.0000) 0.0000(0.0000) | 31.84 0.0005(0.0010) 0.0095(0.0160) | 62.44 0.0005(0.0011) 0.0193(0.0470) | 208.71 0.0015(0.0023) 0.1249(0.1877) |

Table 1: Non-convex problem with $n$ variables and $n$ random linear constraints as $n$ varies as $5, 25, 50, 100$. Results over 100 test instances are listed, using metrics of objective, mean and max inequality constraint violations. The violations are formatted as average value (std) in this Table. Results show that the homotopy heuristics enable a smaller violations than vanilla penalty method.

## 4.2 POWER GRID AC OPTIMAL POWER FLOW PROBLEM

These subsection evaluates on the real-world task of ACOPF problem (defined in Section 2.3). Table 2 shows ACOPF results on a 30-bus system. Homotopy methods improve the feasibility of NN outputs with smaller violations of equality and inequality constraints. See Appendix A for our experiment settings and hyper-parameter tuning.

| Method | Obj | Mean eq. | Max eq. | Mean ineq. | Max ineq. |
|---|---|---|---|---|---|
| **SaS, SBnds** | 666 | 0.0027 (0.0015) | 0.010 (0.005) | 0.0000 (0.0000) | 0.000 (0.001) |
| **SaS, GPen** | 666 | 0.0023 (0.0014) | 0.008 (0.005) | 0.0000 (0.0000) | 0.000 (0.000) |
| **Tx-stepping, SBnds** | 665 | 0.0024 (0.0013) | 0.008 (0.005) | 0.0000 (0.0000) | 0.000 (0.000) |
| **Tx-stepping, GPen** | 665 | 0.0017 (0.0009) | 0.006 (0.003) | 0.0000 (0.0000) | 0.000 (0.000) |
| **Load-stepping, SBnds** | 666 | 0.0020 (0.0012) | 0.007 (0.004) | 0.0000 (0.0000) | 0.000 (0.000) |
| **Load-stepping, GPen** | 667 | 0.0024 (0.0016) | 0.009 (0.005) | 0.0000 (0.0000) | 0.000 (0.000) |
| **vanilla penalty** | 673 | 0.0036 (0.0027) | 0.013 (0.010) | 0.0000 (0.0000) | 0.001 (0.003) |

Table 2: Results of ACOPF problem on case30, over 100 test instances. Mean and max violations are analyzed across the test instances and we list the average value (std). Vanilla penalty method has larger violations of constraints, whereas homotopy methods have smaller violations.

## 5 CONCLUSION

This work proposed the use of homotopy optimization for the unsupervised learning of deep learning models constrained by a large set of (nonlinear) equality and inequality constraints. The homotopy heuristics developed in this paper include general-purpose homotopy heuristics based on relaxation of constraint bounds to enlarge the basin of attraction, as well as physics-informed transformation of domain problem leading to trivial starting points lying within the basin of attraction. Our numerical case studies including a family of abstract and real-world problems indicate that the developed homotopy heuristics achieve a more reliable convergence, giving predictions with improved feasibility on unseen data.

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

## A APPENDIX

### A.1 EXPERIMENT SETTINGS AND HYPER-PARAMETER TUNING

To create a fair comparison in each optimization problem, experiments with and without homotopy heuristics will train with the same NN architecture, Adam optimizer, and learning rate scheduler (StepLR with step=100, gamma=0.1, and a minimal learning rate $10^{-5}$; in homotopy methods, lr scheduler is only applied to the last homotopy step). Early stopping is also applied in each experiment to avoid overfitting: each vanilla method trains for 1,000 epochs with warmup=50, patience=200, and each homotopy method trains 100 epochs in each homotopy step with warmup=50, patience=50. All NNs are trained on PyTorch.

Experiment settings for the non-convex problem with random linear constraints, see problem definition (16) in Section 4.1, are listed below:

- Dataset: 50,000 instances (with train/validation/test ratio 8:1:1)
- NN architecture: cylinder NN with 4 layers, hidden layer size increases with problem size $n$ by $hidden layer size = 30\sqrt{n/5}$
- Penalty weights: $w_{eq} = 50$, $w_{ineq} = 50$
- Homotopy settings: CObj has $f_{cvx} = \sum_{i=1}^{n-1}(1 - x_i)^2 + 2(x_{i+1}^2 + x_i^4)$, SaS has $\epsilon_H = 0.01, \epsilon_L = 0$, SBnds has $\epsilon^+ = 1, \epsilon^- = 0$.

These hyper-parameters are tuned and kept fixed across all experiments of this non-convex problem.

For the ACOPF problem, we propose an additional trick of $P_g$ pull-up and used it on all experiments.

($P_g$ **pull-up**) Based on power system domain knowledge, any decision with supply lower than demand is always technically infeasible. To avoid bad predictions of this type, we apply the heuristics of $P_g$ pull-up where an additional domain-specific constraint $\sum P_g - \sum P_d \geq \epsilon$ is added to pull the generation up and thus promote convergence to a point with total supply higher than demand. This constraint is not subject to homotopy heuristics and remains the same in the homotopy path. experiment settings are as follows. Similarly as for other constraints, we have a weight $w_{pullup}$ to impose the additional constraint in penalty method.

- Dataset: 50,000 instances (with train/validation/test ratio 8:1:1), data are generated by randomly sampling load profiles in the range of $75\% - 150\%$ of the base load profile (base load is the load in case data).
- Batchsize: 1024
- NN architecture: cylinder NN with 2 layers and hidden layer size = 200,
- Penalty weights: $w_{eq} = 10^5$, $w_{ineq} = 10^6$
- General homotopy settings: $\Delta\lambda_H = 0.05$, SaS has $\epsilon_H = 0.01, \epsilon_L = 0$, SBnds has $\epsilon^+ = 1, \epsilon^- = 0$.
- Domain specific homotopy settings: warm homotopy loss has $w_{warm} = 10^6$
- others: $P_g$ pull up has $w_{pullup} = 10^6, \epsilon = 0.01$

The hyper-parameters are kept fixed across all experiments.

