# OpenReview forum: "Homotopy Learning of Parametric Solutions to Constrained Optimization Problems"
_ICLR.cc/2023/Conference — Submitted to ICLR 2023_

### Official Review · Reviewer_Rnhn · 2022-10-15

**Confidence:** 3
**Correctness:** 4
**Technical Novelty And Significance:** 1
**Empirical Novelty And Significance:** 2
**Recommendation:** 3

**Clarity, Quality, Novelty And Reproducibility:**

Clarity: I think the paper writing is OK. I can easily follow all the details.

Quality: The paper quality can be significantly improved. Based on my comments on the weakness of the paper, I think the authors should be a better job on convincing the novelty and significance of their methods. Moreover, they should design much more experiments.

Novelty: I think the novelty of this paper is limited. According to the authors, homotopy is not a new idea in this field; the two types of methods they proposed look pretty straightforward; the experimental results are not significantly better than the previous methods.

Reproducibility: I think reproducibility is not an issue, because there are only two simple experiments in this paper, and the authors provides discussion on experimental settings and hyperparameters.

**Strength And Weaknesses:**

I am not familiar with constraint optimization problems, so my review might be biased.

Strength:
If I understand correctly, the main contribution of this paper is combining homotopy (lambda_H), deep learning (using neural network to do optimization) and diffentiable parametric programing (penalty method) together. And the novelty here is proposing the two types of methods.

Weakness:
I think the homotopy method is intuitive and interesting, but at the current moment, I am not sure how significant the method is. I have the following claims about this paper, but I will be happy to change them after the rebuttal:
1. It seems that combining homotopy, deep learning and DPP is not a very novel idea. According to the related work section, all of these methods were used previously, for the constraint optimization problems.
2. It seems that the proposed methods in Type I are pretty straightforward. Are there anything special about SBnds, GPen, etc? For example, people previously never used such tricks before (which is hard for me to believe), or there are some theoretical supporting evidences behind these methods. Moreover, why do we need epsilon_L and epsilon_H in (12)? The authors may need to provide further clarifications.
3. It seems that there are no general principles for Type II methods, because the authors only provide the methods for power system. If I am solving for another problem, how shall we design new Type II methods?

4. The experimental results do not look convincing.

----4a. I do not really undrstand the numbers here. It seems that all methods achieve pretty good performance, with very small differences.

----4b. The authors are just comparing their method with the penalty method, not other methods. It is hard for me to judge whether their methods are better than other state-of-the-art methods.

----4c. In the related work, the authors mentioned that the existing use of homotopy on classical optimization solvers can be slow. I think it is worthwhile to compare the methods in the paper, with the previous homotopy method, in terms of running time.








**Summary Of The Paper:**

This paper proposes a homotopy learning method for solving constrained optimization problems. The idea of homotopy is pretty intuitive: we have a parameter lambda_h \in [0,1], indicating the difficulty of the current problem. lambda_h=0 means a trivial initial problem H0 and lambda_h=1 means the target difficult problem F. The intermediate problem is defined as H= (1-\lambda_H) H0 + lambda_H F.

Therefore, the main challenge here is to design a good H0, so that the whole optimization process will easy proceed by gradually increasing lambda_h from 0 to 1, and optimize the function H along the way.

Starting from the penalty method (which is classical), the authors introduce two types of methods for designing H0. Type I is general techniques, including CObj, SBnds, GPen, and Sas. Type II is domain-aware techniques, including Load-stepping and Tx-stepping, specifically designed for power system.

The authors conduct two experiments, showing that the proposed techniques can outperform the vanilla penalty method.

**Summary Of The Review:**

I think this paper does not provide convincing evidence supporting the significance of the newly proposed method, both in terms of novelty and experimental results.

---

### Official Review · Reviewer_H3TT · 2022-10-20

**Confidence:** 4
**Correctness:** 2
**Technical Novelty And Significance:** 2
**Empirical Novelty And Significance:** 2
**Recommendation:** 3

**Clarity, Quality, Novelty And Reproducibility:**

The general idea is clear and makes sense.  In terms of novelty it seems to be new in the deep-approximation literature for constraint optimization, but homotopy continuation has been used in a variety of problems.  Execution is poor:  attempts at mathematical rigor fall far short (and are hopeless in such general settings), and experiments are not very convincing -- it's unclear that the computational cost of homotopy (which should be quantified) is worth the small improvement in solutions (objective and constraint violations). Important details are missing.

**Strength And Weaknesses:**

Strength:  the homotopy approach that creates a path of problems connecting hard problems to easy problems is a very established general class of methods, which works well in certain domains.  Approximating optimization problems via neural networks is an interesting and active line of work.  So the idea to use homotopy methods to improve training such neural nets seems sensible.  Also application to power-grid is interesting, and the particular physically-motivated relaxations to make the problem easier are maybe a useful highlight of the paper.

Weaknesses: 1. by considering the fully general problem (non-convex objectives with non-convex equality and inequality constraints) basically you ensure that nothing useful can be said,  as the problem is not solvable.  There are some non-rigorous mathematical musings that if the problem has a single stable isolated global optima, and is "locally convex or Lipschitz" (which is easy to understand for unconstrained optimization, but requires considerably more care for constrained optimization) and for a slow-enough step size starting in the vicinity of the optimum that maybe it should-work.  Rigor is at pictorial level -- this picture of basin-attraction seems to make sense.  In general I'm not sure if there are any results that if the origin problem is convex then following the homotopy will reach some arbitrary local optima of the non-convex one -- if there are such results the authors should cite them explicitly.  To reach global or even good solutions I believe is hopeless without very strong assumptions as homotopy will have various issues -- for example path bifurcations, and path jumps/discontinuities: the mapping from the homotopy parameter to local optima could be highly discontinuous.  Consider a star-shaped constraint set and a linear objective function.  So essentially the method is a heuristic method -- which could in practice be useful for some problems -- if the results were very thorough and convincing.

2.  The scope is unclear, and connection to parametric programming is poorly described.  The authors come up with yet another name for the problem 'differentiable parametric programming' -- but do not really explain the link to parametric programming.  E.g. in parametric LP there are usually 1 (or a small number) of parameters that can vary, and the others are fixed. The goal is to characterize how the optimal solution varies w.r.t. to these small set of parameters.   In your description it seems it could range from 1 parameter to all parameters?  Do you think your method is able to solve say arbitrary QPs via neural nets,  or is it limited to small-dimensional parameters?  For 1D parametric programming it may be cheaper to have a grid of the parameter, and solve the optimization problem, and interpolate, avoiding expensive training of DNNs.  For convex parametric programming, the solution path can be characterized (e.g. parametric LPs and some QPs have piecewise linear solution paths).   It's unclear that the power-grid problem  has any structure that makes the problem easier to solve than generic non-convex constrained problems.

3. Use of very informal language and unsupported claims, while attempting to appear mathematically rigorous. a) What does "reaching reliable convergence" mean?  Do you mean converging to a local optimum without constraint violations? Such terms used throughout the paper have to be clearly defined. (b) "There's a high-likelihood that the minimizer will be in basic of attraction".  Is this fully informal, or is there some probabilistic model maybe?  (c)   pNLP methods are NP-hard.  non-convex problems are in general not efficiently solvable.  Being parametric doesn't not itself create the problem.  E.g. 1-dimentional parametric LPs are efficiently solvable.
(d) "Penalty methods have no guarantees of satisfying constraints" -- there's existing literature on "exact penalty methods" started by Prof. Bertsekas,  that do have rigorous guarantees in the convex setting.
"Necessary and sufficient conditions for a penalty method to be exact", Dimitri P. Bertsekas. Mathematical Programming volume 9, pages 87–99 (1975) . For non-convex setting no guarantees are possible essentially for any methods.
(e) Smaller violation of constraints in real world means more practical solution. "Better feasibility"...  in some settings it's binary: either feasible or infeasible.  (f) "Studies have shown that" -- is used a few times.  It's unclear if this is experimental studies or formal mathematical argument -- please say, and provide references.

4. No discussion of how to select your penalty weights w.  It seems pretty important.   No discussion of feasibility.  How do you know if some parameter setting is even feasible?

5. Weak experiments.   The results are not well explained -- e.g. in Table 1 and Table 2 it's not clear what the different methods are, which are homotopy methods and which aren't.   Details of the experiments are missing (some are in the appendix), but others are not discussed -- e.g. how to set w-weights (penalty weights).   Results are mostly incremental increases (I'm assuming the bottom case 'vanilla' is non-homotopy). The others are incrementally better -- especially for the power-grid problem.  Making constraint violations an order of magnitude smaller may be qualitatively different, but significance of small improvements   (0.13 to 0.10) -- I'm not clear about.


**Summary Of The Paper:**

The paper considers approximating solutions of constrained optimization problems using neural networks.  A neural network is trained to approximate a parametric family of such problems, mapping the parameters to the optimal solution (i.e. arg-max).  The objective for training the neural net is a combination of the objective of optimization plus constraint violations scaled by some weight (penalty method).  The innovation in the paper is to consider a homotopy of optimization problems -- i.e. connecting a continuum of optimization problems interpolating between the hard problem one ultimately wants to solve and an easy problem, and attempting to follow a solution path.  The setting is a fully general non-convex optimization with non-convex equality and inequality constraints (which is essentially hopeless in its full generality). An application to power-grid is considered.

**Summary Of The Review:**

The idea of using homotopy continuation to train neural nets to approximate solutions of constraint optimization problems is sensible.  Experiments are weak, and while the do show evidence that homotopy may be marginally helpful -- they are not very convincing.  Very informal language, including problem definition, and important missing detail.  If you do attempt formal arguments supporting homotopy methods -- you may need to limit the scope, impose clear assumptions, and cite specific results in existing literature in convex optimization / homotopy methods. Othwerwise it appears to be loosely-motivated intuitive mathematical musings, rather than formal argument.

---

### Official Review · Reviewer_JJRB · 2022-10-26

**Confidence:** 3
**Correctness:** 3
**Technical Novelty And Significance:** 2
**Empirical Novelty And Significance:** 2
**Recommendation:** 5

**Clarity, Quality, Novelty And Reproducibility:**


I appreciate the authors' effort in clarifying the details heuristics as clear as possible. However, the paper still appears to be improvable

It appears that convexifying the objective is critical for the empirical performance, and can you elaborate how to choose $f_{cvx}$?
My feeling is that a poor choice of $f_{cvx}$, can make the initial solution is far from the real optimal one $x^*$, which makes it even more costly to use homotopy approach.

- In page 8, it is a bit strange to call a problem (4) the vanilla method.  Is it about training the network with loss function $L$ above directly?
Is it possible to provide the results solved using the solver  as baseline?
- What is the special meaning of these experimental results in red color?
- The data size of 30 bus appears to be a small scale problem. Can you justify your results on some larger datasets? For example, the one [here](https://github.com/power-grid-lib/pglib-opf).

Not sure about reproducibility. Maybe I missed something, but I didn't find any source code.

**Strength And Weaknesses:**

The paper is well-written and easy to follow. It clearly illustrated the intuition of the homotopy method and the empirical performance.

Weakness
The soundness of the proposed homotopy method is not fully convincing. While I can see the intuition from the paper, it is hard to convince me that one can easily find the right hyper-parameters to efficiently use homotopy method.

Moreover, I think the paper lacks comparison with many other important algorithms for constrained optimization, such as augmented Lagrangian and primal-dual method, which all have theoretically justifiable ways to tune the penalty weight.


**Summary Of The Paper:**

The paper proposes a new deep learning approach for solving constrained optimization with the help of  homotopy heuristics. The key idea is to create a continuous transformation of objective and constraint with gradually increasing complexity. While there is no theoretical analysis provide, the experimental results show that the heuristics can greatly improve the performance of penalty methods  on both  synthetic problem and the real-world AC optimal power flow problem.

**Summary Of The Review:**

Overall, this paper provides an interesting idea to make learning based method more effective in constrained optimization. However, as far as I know, penalty method is as practical as Lagrangian-based methods, as the Lagrangian method  can automatically adjust the penalty (Lagrangian multipliers). For a fair comparison, I suggest the author compares with the state-of-the-art algorithm in these areas. I also have some concerns about the soundness of the algorithm as stated above.

---

### Decision · Program_Chairs · 2023-01-20

**Decision:**

Reject

**Justification For Why Not Higher Score:**

See problems above, not addressed as the author response is missing

**Justification For Why Not Lower Score:**

n/a

**Metareview: Summary, Strengths And Weaknesses:**

This paper proposes homotopy approaches toward solving a broad class of constrained problems. The techniques are sophisticated which demands that they are supported rigorously, yet reviewers found many claims to be informal and imprecise. The claims are not all supported, and several reviewers found that it was difficult to identify the methodological core of the paper, though the authors seem to attempt to combine several good ideas from the existing literature. A common opinion was that the experimental validation does not sufficiently support the claims.